# A Hybrid Technique for Diabetic Retinopathy Detection Based on Ensemble-Optimized CNN and Texture Features

**DOI:** 10.3390/diagnostics13101816

**Published:** 2023-05-22

**Authors:** Uzair Ishtiaq, Erma Rahayu Mohd Faizal Abdullah, Zubair Ishtiaque

**Affiliations:** 1Department of Artificial Intelligence, Faculty of Computer Science and Information Technology, University of Malaya, Kuala Lumpur 50603, Malaysia; 2Department of Computer Science, COMSATS University Islamabad, Vehari Campus, Vehari 61100, Pakistan; 3Department of Analytical, Biopharmaceutical and Medical Sciences, Atlantic Technological University, H91 T8NW Galway, Ireland; zubair.ishtiaque@research.atu.ie

**Keywords:** machine learning, deep learning, feature engineering, diabetic retinopathy, medical imaging, artificial intelligence techniques

## Abstract

One of the most prevalent chronic conditions that can result in permanent vision loss is diabetic retinopathy (DR). Diabetic retinopathy occurs in five stages: no DR, and mild, moderate, severe, and proliferative DR. The early detection of DR is essential for preventing vision loss in diabetic patients. In this paper, we propose a method for the detection and classification of DR stages to determine whether patients are in any of the non-proliferative stages or in the proliferative stage. The hybrid approach based on image preprocessing and ensemble features is the foundation of the proposed classification method. We created a convolutional neural network (CNN) model from scratch for this study. Combining Local Binary Patterns (LBP) and deep learning features resulted in the creation of the ensemble features vector, which was then optimized using the Binary Dragonfly Algorithm (BDA) and the Sine Cosine Algorithm (SCA). Moreover, this optimized feature vector was fed to the machine learning classifiers. The SVM classifier achieved the highest classification accuracy of 98.85% on a publicly available dataset, i.e., Kaggle EyePACS. Rigorous testing and comparisons with state-of-the-art approaches in the literature indicate the effectiveness of the proposed methodology.

## 1. Introduction

The retina is a spherical structure present at the back of the eye. Its function is to process visual information through specialized cells present within it, known as rods and cones. The retina receives its blood supply through the eye’s vascular system. In addition, it requires an unobstructed blood supply and a highly maintained blood sugar level for optimal function [1]. However, in patients with uncontrolled diabetic mellitus, large amounts of sugars start accumulating in the blood, which leads to the breakdown of blood vessels due to improper distribution of oxygen to the cells, leading to structural abnormalities in the blood vessels that eventually cause diabetic retinopathy (DR) [2]. Diabetic retinopathy is a very common condition that occurs in patients who suffer from diabetes mellitus. It is the most common cause of adult blindness worldwide. In the United States, approximately 4.1 million people over the age of 40 suffer from DR. One in every twelve people of this age is reported to suffer from severe vision-threatening retinopathy [3].

The major signs of DR include microaneurysms (MA), exudates (EX), hemorrhages (HE), and cotton wool spots (CWS). Moreover, the major symptoms of DR include swelling of the blood vessels, floating, flashing, blurry vision, and sudden vision loss [4]. Moreover, diabetic retinopathy has two stages: non-proliferative diabetic retinopathy (NPDR) and proliferative diabetic retinopathy (PDR). NPDR is further divided into mild, moderate, and severe DR depending upon the severity of the condition. During this stage of DR, the blood vessels in the eye rupture, and fluid begins to leak into the retina. Consequently, the retina becomes swollen and wet, making it less sensitive [5]. Furthermore, MAs are the primary indicators of NPDR. When the major signs of DR (MA, EX, HE, and CWS) worsen, it causes neovascularization, which consequently results in PDR [6,7]. It is sometimes thought that MA and HE belong to the same class of red lesions [8]. Nonetheless, the detection of DR at an early stage is very important in order to save patients’ vision.

Automated DR detection is an area of intense research since it is related to healthcare. Many researchers believe that 90% of DR patients’ vision can be saved if it is diagnosed at an early stage [9]. Moreover, automated DR detection has several benefits over manual DR detection, which include a reduced chance of human error, a lower workload for ophthalmologists, and the extraction of even the most minor lesions that hard for ophthalmologists to accurately recognize. Many researchers have conducted their research in the area of automated diabetic retinopathy detection. These studies utilize different image preprocessing techniques, including green channel extraction [10], contrast enhancement through contrast-limited adaptive histogram equalization (CLAHE) method [11], optical disc and blood vessel removal [12], and resizing [13] for the region of interest (ROI). Moreover, after image preprocessing, classification can be performed using artificial intelligence techniques (either machine learning (ML)-based or deep learning (DL)-based models). In ML models, first, discriminating features are extracted from the preprocessed fundus image. These features may include statistical features [14], color features [15], intensity features [16], shape and structure features [17], and texture features [18]. Furthermore, these extracted features are used to build a master feature vector (MVF). Then, this MVF is fed to the machine learning (ML) classifiers [19] to construct a diabetic retinopathy classification model. However, in deep learning (DL) models, the extraction of discriminative features automatically takes place from the training data, and then, classification of the test data into different diabetic retinopathy classes is executed. The major challenge in DR detection is the extraction of distinctive features from the retinal images, which can be improved using ML classifiers [20].

In this article, a hybrid methodology is proposed for diabetic retinopathy detection and classification. For the accurate classification of DR abnormalities, we utilized dataset preprocessing algorithms to make subtle information more prominent. We utilized image enhancement algorithms for the improvement of image quality. After the preprocessing step, we performed feature engineering comprising three major steps: feature extraction, feature selection, and feature fusion. In this research, we designed a deep learning model named GraphNet124 and utilized a modified version of the ResNet50 model for feature extraction from Kaggle EyePACS [21], which is a publicly available dataset. The major contributions of this paper are as follows:We propose an efficient hybrid technique that uses an ensemble-optimized CNN for automated diabetic retinopathy detection to improve classification accuracy.We propose a novel GraphNet124 for feature extraction and to train a pretrained ResNet50 for diabetic retinopathy images, and then, the features are extracted using the transfer learning technique.We propose a feature fusion and selection approach that works in three steps: (i) features from GraphNet124 and ResNet50 are selected using Shannon Entropy, and then, fused; (ii) these fused features are optimized using the Binary Dragonfly Algorithm (BDA) [22] and the Sine Cosine Algorithm (SCA) [23]; (iii) the features extracted from LBP are also selected using Shannon Entropy, and then, fused with the optimized features found in step (ii).We evaluate the proposed hybrid architecture on a complex, publicly available, and standardized dataset (Kaggle EyePACS).We compare the performance of the proposed hybrid technique, including the fusion of discriminative features from GraphNet124, ResNet50, and LBP, with baseline techniques.To the best of our knowledge, this study is the first in the domain of DR abnormality detection and classification using the fusion of automated CNN-based features and LBP-based textural features.

The rest of the paper is organized as follows. Section 2 describes related works on diabetic retinopathy detection. Section 3 states the proposed methodology, including the dataset used to perform the experiments, the image preprocessing techniques used in the study, and the proposed feature engineering methods: LBP feature extraction, CNN feature extraction, feature selection, and fusion. Section 4 presents the results of different experimental setups applied using different performance measures. Finally, Section 5 concludes this paper.

## 2. Related Work

Over the past few years, researchers have rapidly contributed to the area of medical image processing for medical abnormality recognition and classification. The utilization of advanced machine learning and deep learning models has revolutionized research outcomes. In medical imaging, the predominant areas where researchers are contributing and utilizing advanced image processing and computer vision algorithms are stomach abnormality detection, brain tumor detection, skin lesion detection, breast cancer detection, and diabetic retinopathy detection. Scientists have proposed a variety of techniques for categorizing diabetic retinopathy using colored fundus images [24].

The authors of [25] proposed a method for retinal lesion classification. A genetic algorithm was utilized in their research based on an optimal weight learning process for each classifier. Though the feature set used in their model made it complex, their method correctly classified the retinal images into NPDR classes. Luo et al. [26] proposed a self-supervised fuzzy clustering network (SFCN) to deal with the problem of the manual annotation of retinal dataset images. Their method achieved better performance to circumvent the difficulty of manually annotating a huge number of retinal pictures. The performance of the SFCN approach was satisfactory, but there is still potential for the development of a DR detection and classification technique that can outperform the existing supervised learning methods. Vijayan, Sangeetha [19] extracted their features using a Gabor filter for the detection of diabetic retinopathy. They achieved a maximum accuracy of 70.15% on a subset of a Kaggle-EyePACS dataset (10,024 images) using a Random Forest (FR) classifier. A multi-tasking deep learning method was proposed by Wang et al. [27], for the contemporaneous diagnosis of diabetic retinopathy severity level. The proposed hierarchal structure incorporated the relationships among the diabetic retinopathy features and levels of severity. In a study proposed by Ali Shah et al. [28], MA was detected by utilizing Hessian, curvelet-based, and color feature extraction and achieved a sensitivity of 48.2%.

Orlando et al. [29] proposed a method for red spot detection. They proposed a hybrid method based on color equalization with CLAHE-based contrast enhancement, handcrafted features, and CNN features. For lesion classification, the Random Forest Classifier was used and achieved an AUC of 0.93. Bhardwaj, Jain [30] presented a hierarchical severity level DR grading system using two publicly available datasets: MESSIDOR and IDRiD. The authors extracted shape, intensity, and GLCM (Gray-Level Co-occurrence Matrix) textural features, which were then fed to the KNN (K-Nearest Neighbor) and SVM (Support Vector Machine) algorithms and achieved accuracy levels of 95.30% and 92.60%, respectively, on the MESSIDOR dataset. For the IDRiD dataset, they achieved 94.00% accuracy using the KNN classifier. Nonetheless, their proposed approach may be inefficient for a large and complex dataset such as Kaggle-EyePACS. In [31], the authors proposed a method for the detection of diabetic retinopathy from the sub-images (patches) of the dataset. They utilized state-of-the-art deep learning models such as VGG16, GoogleNet, AlexNet, and ResNet. A small subset of the Kaggle-EyePACS dataset was utilized in their research, comprising only 243 images. Their method achieved an accuracy of 98.0%.

Keerthiveena et al. [32] proposed a method for the early-stage detection of diabetic retinopathy and glaucoma detection. Their method was based on three major phases: preprocessing, feature selection, and classification. In the first phase, the green channel was extracted from the fundus images and was further improved using the CLAHE method. This method achieved 98.20% accuracy with 10-fold cross-validation. Additionally, in research conducted by Zhao et al. [33], they proposed a method for retinal vessel segmentation using region growing and level set method. The images were preprocessed using the CLAHE, and a 2D Gabor wavelet filter was applied to improve the vessel’s quality. The dataset was smoothened to preserve the boundary information of the blood vessels. Retinal vessels were segmented using a hybrid technique consisting of the region-growing and region-based active contour methods with the implementation of a level set. In [34], five machine learning models were utilized by the authors on a private dataset. They used k-means clustering to segment the lesions. Using the segmented image, they extracted the features using wavelet, grayscale co-occurrence run-length matrices, and histograms. Nevertheless, when comparing this procedure to current state-of-the-art techniques, the highest accuracy produced by their experiments was 99.73%. Since they used a private dataset, their results could be biased. Additionally, when they applied their approach to a subset (100 DR images) of a public dataset (Messidor), they achieved 98.83% accuracy. Moreover, their approach may be ineffective for a larger dataset such as Kaggle-EyePACS.

In this section, we discussed the state-of-the-art methods of diabetic retinopathy detection and classification. It is observed from the discussed literature that studies utilizing deep learning methods achieved significant detection performance when applied to larger datasets. Impressed by the state-of-the-art methods, we propose a hybrid technique for determining DR grade or performing severity level categorization, which is described in the following section.

## 3. Proposed Methodology

We propose a hybrid method in this research methodology to identify and classify various retinal abnormalities. The five major steps of our method are as follows: dataset preprocessing, feature extraction using deep learning models, feature selection, and feature optimization, as well as classification using machine learning algorithms. Figure 1 depicts a block diagram of the proposed method.

Our proposed method can be described in the following phases:

*Dataset:* An online public dataset, the Kaggle-EyePACS dataset [21], was used for the detection and classification of DR images into specific classes.

*Preprocessing:* The DR images were then preprocessed, since preprocessing is an important phase in DR detection and classification. The preprocessing steps followed in this study were image resizing, data augmentation, applying a median filter, and image sharpening.

*Feature Engineering:* Distinguishing features were then extracted and selected from the preprocessed dataset. Three methods were used for feature extraction, namely, Local Binary Patterns (LBP) for texture-oriented features, and the novel GraphNet124 and ResNet50 for CNN-based features.

*Feature Selection and Fusion:* After feature extraction, salient features from LBP, GraphNet124, and ResNet50 were selected using the Shannon Entropy algorithm. Moreover, these selected features were then fused and optimized using the Binary Dragonfly Algorithm (BDA) [22] and the Sine Cosine Algorithm (SCA) [23].

*Classification and Evaluation:* The optimized feature vector was fed to ten ML algorithms, including five variants on SVM and five variants of KNN, for the classification of DR images into five severity classes. Finally, these algorithms were evaluated using different evaluation matrices, namely, specificity (SPE), F1-Score (F1), accuracy (ACC), precision (PRE), sensitivity (SEN), and time (in seconds).

The details of the aforementioned phases are discussed in the subsequent sections.

### 3.1. Dataset

In this research study, we utilized the “KAGGLE Diabetic Retinopathy Detection” EyePACS dataset [21]. Each image showed different diabetic retinopathy lesions (including MAs, EX, CWS, and HE), graded by a medical professional using the following scale: no DR (Class 0), mild (Class 1), moderate (Class 2), severe (Class 3), and proliferative DR (Class 4). Different camera models and setups were used to collect the photos in the dataset, which could have affected the quality of the retinal images. This is the largest publicly available dataset of DR images. However, a large number of images in this database contain noise. For instance, some images are blurred, and some others are over-exposed. This dataset comprises 35,126 training images of the mentioned classes.

In this research study, we utilized the complete dataset for the deep learning model training. However, for feature extraction, we used a total of 15,000 images with 3000 images in each class. We utilized data augmentation techniques to balance the dataset. Figure 2 shows some of the sample images of the Kaggle-EyePACS.

### 3.2. Preprocessing

The first phase of the proposed model was dataset preprocessing. In this phase, we improved the quality of the images in four steps. The dataset was originally in different dimensions. To standardize them, we resized the dataset images to 512 × 512. After the resizing step, we used a data augmentation technique to balance the data, since Kaggle EyePACS is an imbalanced dataset and the results can be biased. Some augmented sample images are given in Figure 3. After the augmentation step, we applied a median filter to the entire dataset for noise removal from the images, since a median filter is an image smoothing technique and it retains the edges while removing noise. Figure 3 shows the effect of median filtering on the resized image. In the third step of preprocessing, we utilized an unsharp-masking filter to improve the contrast of the image and highlight the edges to sharpen them. By first making a blurred version of the original image, and then, subtracting it from the actual image, the sharpening filter worked. The outcome was an image with a high-pass filter that highlights the original image’s edges with finer features. By boosting the contrast between the image’s edges and details, this procedure improved the visual quality of the images. To optimize the outcomes for DR, the filter’s parameters, including its size, the degree of blurring, and the strength of its sharpening effect, were changed. The effect of image sharpening is depicted in Figure 3.

### 3.3. Feature Engineering

Feature extraction and engineering were the most important steps of our proposed method, as these steps affect the method’s performance. Appropriate feature extraction was the most critical task. An overview of the suggested feature engineering technique is provided in this section. Moreover, to detect and categorize the DR grades, we extracted texture characteristics and deep learning features in this research. The following subsections provide a brief explanation of the feature engineering phase of the suggested method.

#### 3.3.1. LBP Feature Extraction

We extracted the Local Binary Patterns (LBP) for texture-oriented features. LBP is an important technique for locating and identifying objects. LBP features are two bitwise transitions from 0 to 1 and 1 to 0, respectively. LBP calculates the mean and variance for each pixel’s intensity using a greyscale image as its input. The following formulation is used to represent LBP mathematically:(1)$$Texture\;Features_{\mathrm{LBP}}\left(T,R\right)=\overset{T-1}{\underset{T=0}{\sum }}S\left(U_{T}-U_{ℭ}\right)2^{T}$$

Here, T is the number of neighborhood intensities, R denotes the radius, $U_{T}$ denotes the variance of the nearby pixel intensity, and $U_{ℭ}$ denotes the intensity contrast determined from (T, R).
(2)$$S_{𝓃}\left(T\right)=\left\{\begin{array}{c} 1\;\;\;\;\;\;\;\;\;\;\;\;\;\;\;\;\;if\;T\geq t\\ \;\;\;0\;\;\;\;\;\;\;\;\;\;\;\;\;\;\;Otherwise \end{array}\right.$$
where the central pixel “t” is compared to the surrounding pixels $S_{𝓃}\left(T\right)$. It generates a feature vector with dimensions of 1 × 59 for a single image and N × 59 for N images.

#### 3.3.2. CNN Feature Extraction

Deep learning features were extracted using the proposed CNN model and the pre-trained ResNet50 architecture. In this research work, we proposed a deep learning model for the classification of the DR dataset. The proposed model was designed in a branching layout. This proposed model is named GraphNet124, since it contains a total of 124 layers. We pre-trained the proposed deep learning model on the CIFAR-100 dataset, and upon later using the transfer learning technique, it was trained on the 50,000 images of the Kaggle-EyePACS dataset. Details of the dataset are provided in the Dataset section. The layered architecture of the proposed GraphNet124 is given in Figure 4.

In the feature extraction step, we extracted two types of deep CNN features, and texture features were obtained using LBP. After this step, we obtained two feature vectors with dimensions of 15,000 × 4096 and 15,000 × 2048 from the proposed GraphNet124 and ResNet50 CNN models, respectively. Moreover, training of our deep neural network was performed using the process of fine-tuning the hyperparameters. We trained the model using an SGDM (Stochastic Gradient Descent with Momentum) optimizer with a validation frequency of 50, and the maximum epochs used for the training were 50 and 100 for 5-fold and 10-fold cross-validation experiments, respectively, with a minibatch size of 64. Furthermore, we utilized an L2 regularization of 0.0001 and shuffled images at every epoch, with the learning rate dropped by a factor of 0.1.

#### 3.3.3. Feature Selection and Fusion

Feature selection was performed using the Shannon Entropy algorithm. The feature selection was conducted using a heuristic method. Both vectors were independently used to calculate the Shannon Entropy, and the target function was defined depending on the average value of the original entropy vectors. Machine learning classifiers were fed with robust features, which were those that were either equal to or better than the mean features. However, this procedure must continue until the classifier’s error rate is less than 0.1. Shannon Entropy is mathematically supported by the following equation:

Where $o_{i}^{k}$ represents the total number of occurrences of $r_{i}$ in the class or category $C_{k}$, and $r{𝒻}_{i}^{k}$ denotes the frequency of the $r_{i}$ in the category $C_{k}$:(3)r𝒻ik=oik ∑k oik

Whereas the Shannon Entropy E($r_{i})$ of the term $r_{i}$ is mathematically formulated as:(4)$$E\left(r_{i}\right)=-\overset{t}{\underset{k=1}{\sum }}\left(r{𝒻}_{i}^{k}\right)\times \left(log_{2}\left(r{𝒻}_{i}^{k}\right)\right)$$

After the selection of features, we obtained the feature vectors $F_{LBP}$, $F_{GraphNet124}$, and $F_{ResNet50}$ with dimensions of 15,000×l, 15,000×g, and 15,000×r, respectively. Here, *l*, *g*, and *r* represent the total number of selected features obtained for $F_{LBP}$, $F_{GraphNet124},$ and $F_{ResNet50}$, respectively, for all the images of the dataset. These features are defined on the ω sample space and the selected features are the samples, such that ξ ϵ ω. After the feature selection step, we fused the selected features, where the fused feature vector is represented by $E=F_{GraphNet124}F_{ResNet50}$, with dimensions of $\left[15,000\times \left(g+r\right)\right]$. This fused feature vector was optimized using the Binary Dragonfly Algorithm (BDA) [22] and the Sine Cosine Algorithm (SCA) [23], and was named E𝒻𝓋. The output vector was then ensembled with the extracted texture feature vector. The final fused feature vector, named $F_{E𝒻𝓋\left(Final\right)}$, was supplied to the classifiers.

## 4. Results and Discussion

For the classification of DR anomalies in this research, we used ten machine learning methods. For the detection and categorization of DR severity levels, we used two important machine learning techniques: SVM and KNN. The classifiers utilized in this research work are listed in Figure 5. In this section, a detailed discussion on the experimental setup, dataset, and performance measures, and a comprehensive analysis of results, are given.

### 4.1. Experimental Setup

The experimental setup of the proposed method is discussed in this section. This research work is categorized into two main categories (the detection and classification of DR) using an ensemble feature vector with dimensions of 15,000 × 1030 and 15,000 × 2030 with 5-fold and 10-fold cross-validation, respectively. The experiments for detection and classification were performed on a system with 16 GB RAM and a 3.40 GHZ processor. The subsequent sections discuss the dataset utilized, the performance measures considered, and the results of the experiments performed in detail.

### 4.2. Dataset

In this research work, we performed the detection and classification of DR grades. For this purpose, we utilized the “Kaggle EyePACS dataset” [21]. This dataset consists of five grades or classes of DR that are numbered from 0 to 4. These classes contain images for normal, mild, moderate, severe, and proliferate DR. We considered 50,000 images for the training of the proposed CNN model and ResNet50 model. After training the model, we utilized 15,000 images for validation of the proposed technique (consisting of 3000 augmented images in each class). A ratio of 70:30 was used in this research work, where 70% of the DR images were used for training and 30% images were used for testing our proposed method.

### 4.3. Performance Measures

The ensemble feature vector was used to assess the effectiveness of the suggested classification strategy. Specificity (*SPE*), F1-Score (*F1*), accuracy (*ACC*), precision (*PRE*), sensitivity (*SEN*), and time were the performance metrics considered for the classification procedure (*seconds*). The mathematical formulation of these matrices is given as follows:(5)SEN=TPTP+FN
(6)SPE=TNFP+TN
(7)PRE=TPTP+FP
(8)ACC=TP+TNTP+TN+FP+FN
where *TP* = true positive, *FN* = false negative, *TN* = true negative, and *FP* = false positive.

### 4.4. Experiment 1: Classification Results Using Feature Vector with Dimensions of $E\left(15,000\times 1000\right)$ and 5-Fold Cross-Validation

In the first experimental setup, we performed experiments on the fused feature vector and utilized 15,000 images from the Kaggle EyePACS dataset. Whereas the proposed CNN model was initially trained on the CIFAR-100 dataset, later, we utilized the transfer learning technique for post-training of the model on 50,000 images of the balanced Kaggle EyePACS dataset. In this experiment, a feature vector with dimensions of 15,000 × 4096 was extracted from the FC-1 layer of the proposed CNN model. A feature vector with dimensions of 15,000 × 2048 was obtained from the ResNet50 architecture. Meanwhile, texture features were extracted using the LBP algorithm. In the next step, feature selection was performed using Shannon Entropy. After the selection of features, we obtained the feature vectors $F_{LBP}$, $F_{GraphNet124}$, and $F_{ResNet50}\;$ with dimensions of 15,000 × 30, 15,000 × 500, and 15,000 × 500, respectively. Moreover, for the detection and categorization of DR severity levels, we used two important machine learning techniques, namely, SVM and KNN.

For the evaluation of the proposed method, we utilized the ensemble feature vector $E{\left(15,000\times 1000\right)}_{\;}$, obtained after the fusion of the selected features ($F_{GraphNet124\;}F_{ResNet50})$. This feature vector was supplied to the BDA and SCA algorithms for optimization, which resulted in optimized feature vectors. In this experiment, the optimization algorithms were trained with 100 epochs. The optimized feature vectors were fused with the extracted texture features and fed to the SVM and KNN classifiers to evaluate the performance of the proposed technique. The class-wise results for the classification of DR abnormalities achieved using SVM classifiers are given in Table 1.

The final optimized feature vector was supplied to the KNN classifiers so that they could assess how well the proposed strategy worked using different KNN machine learning algorithms. In this experiment, 5-fold cross-validation and a ratio of 70:30 for the training and testing were used, respectively. The class-wise numerical findings for the KNN classifiers’ classification of DR abnormalities are shown in Table 2. In this experiment, the Medium KNN classifier achieved the highest classification accuracy of 95.75%.

### 4.5. Experiment 2: Classification Results Using Feature Vector with Dimensions of $E\left(15,000\times 2000\right)$ and 5-Fold Cross-Validation

This experiment was also performed on a total of 15,000 images from the Kaggle EyePACS dataset. In this experiment, a feature vector with dimensions of 15,000 × 4096 was extracted from the FC-1 layer of the proposed CNN model, i.e., GraphNet124. Similarly, a feature vector with dimensions of 15,000 × 2048 was obtained from the ResNet50 architecture. In addition, texture features were extracted using the Local Binary Patterns algorithm. Afterwards, feature selection was performed through Shannon Entropy. We obtained the feature vectors $F_{LBP}$, $F_{GraphNet124}$, and $F_{ResNet50}$ with dimensions of 15,000 × 30, 15,000 × 1000, and 15,000 × 1000 for LBP, GraphNet124, and ResNet50, respectively. The ensemble feature vector E(15,000 × 2000) was obtained after the fusion of the selected features ($F_{GraphNet124}F_{ResNet50})$.This feature vector was supplied to the BDA and SCA algorithms for optimization, which resulted in optimized feature vectors. Moreover, the optimization algorithms were trained with 50 epochs. The optimized feature vectors and texture features were fused and fed to the SVM and KNN classifiers for evaluation. The results for the classification of DR abnormalities using 2000 features, which were later optimized using the BDA and SCA algorithms with SVM classifiers, are given in Table 3.

The KNN classifiers were fed with the final optimized feature vector so that they could evaluate how well the proposed method performed. In this experimental setup, 5-fold cross-validations were performed, and 70% of the images were used for training and 30% images were used for testing. The results of Experiment 2 using the KNN classifier are given in Table 4. From the results of Experiment 2, the Quadratic SVM achieved the maximum accuracy of 98.35% compared to the other types of KNN classifier.

### 4.6. Experiment 3: Classification Results Using Feature Vector with Dimensions of $E\left(15,000\times 1000\right)$ and 10-Fold Cross-Validation

This experiment was also performed on a total of 15,000 images. In this experiment, a feature vector with dimensions of 15,000 × 4096 was extracted from the FC-1 layer of the proposed GraphNet124 model. On the other hand, a feature vector with dimensions of 15,000 × 2048 was obtained from the ResNet50 architecture, and texture features were extracted using the LBP algorithm. After feature extraction, the next important step was feature selection, which was performed using Shannon Entropy. Feature selection resulted in obtaining three feature vectors, namely $F_{LBP}$, $F_{GraphNet124}$, and $F_{ResNet50}$, with dimensions of 15,000 × 30, 15,000 × 500, and 15,000 × 500, respectively. For the evaluation of Experiment 3, we used the ensemble feature vector E(15,000 × 1000), obtained after the fusion of the selected features $\left(F_{GraphNet124}F_{ResNet50}\right)$. The BDA and SCA algorithms optimized the fused feature vector. In this experimental setup, the optimization algorithms were trained with 100 epochs. Moreover, the optimized feature vectors were supplied to SVM and KNN classifiers, along with the retrieved texture data, for the classification of DR grades. The class-wise results for the classification of DR grades achieved using 1030 features, and afterwards, optimized using the BDA and SCA algorithms with SVM classifiers, are given in Table 5.

To assess the results, we employed several performance indicators, including ACC, SEN, SPE, PRE, F1, and time. In addition, five KNN classifier variants—Fine KNN, Medium KNN, Coarse KNN, Cubic KNN, and Weighted KNN—were used in this experiment. The resulting optimized feature vector was fed to the KNN classifiers for classification. Nevertheless, in this experiment, 10-fold cross-validation and a ratio of 70:30 for the training and testing were used, respectively. The results of this experiment are given in Table 5 and Table 6 for the SVM and KNN classifiers, respectively. The results clearly show the superiority of the Quadratic SVM over all other classifiers as it attained 98.85% accuracy.

### 4.7. Experiment 4: Classification Results Using Feature Vector with Dimensions of $E\left(15,000\times 2000\right)$ and 10-Fold Cross-Validation

This experiment was also performed on a total of 15,000 images from the Kaggle EyePACS dataset. First, a feature vector with dimensions of 15,000×4096 was extracted from the FC-1 layer of the proposed CNN model (GraphNet124). Second, a feature vector with dimensions of 15,000×2048 was obtained from the ResNet50 architecture. Third, texture features were extracted using the LBP algorithm. Then, feature selection was performed using Shannon Entropy, which resulted in three feature vectors, namely $F_{LBP}$, $F_{GraphNet124}$, and $F_{ResNet50}$, with dimensions of 15,000×30, 15,000×1000, and 15,000×1000, respectively. For the evaluation of Experiment 4, we utilized the ensemble feature vector $E{\left(15,000\times 2030\right)}_{\;}$, obtained after the fusion of the selected features ($F_{LBP}F_{GraphNet124}F_{ResNet50})$. This feature vector was supplied to the BDA and SCA algorithms for optimization, which resulted in optimized feature vectors. In this experiment, optimization algorithms were trained with 100 epochs. For the classification of DR grades, the optimized feature vectors were supplied to different variations of the SVM and KNN classifiers, along with the retrieved texture data, to assess their performance.

The classification results of the DR grades, achieved using the 2030 feature with SVM and KNN classifiers, are given in Table 7 and Table 8, respectively. From the analysis of these results, it is observed that the Quadratic SVM achieved promising results compared to the other classifiers. The maximum accuracy achieved in this experiment was 98.41% using the Quadratic SVM.

### 4.8. Comparison with Existing Methods

In this study, we compared the proposed method in terms of ACC, SEN, SPE, PRE, and F1 score. A comparison of the existing methods is given in Table 9. The authors of [35] proposed a method for DR recognition, named the MXception model, for the Kaggle EyePACS dataset. They trained their model on a subset of the dataset comprising a total 19,316 images (10,000 Class 0, 2443 Class 1, 5292 Class 2, 873 Class 3, and 708 Class 4). To make the dataset balanced, they used a class weighting method. Moreover, they employed a pretrained Xception model by chopping the last fully connected layer, and added an average pooling layer and a one-neuron dense layer as their output layer. This model achieved a promising accuracy of 82%. Li et al. [36] trained two Deep CNNs and replaced their traditional max pooling layers with fractional pooling layers while utilizing the Kaggle EyePACS dataset, where they used 34,124 images to train the network and 1000 images to validate their model, and finally, performed the testing with 53,572 images. Their proposed method achieved 86.17% accuracy, which was better than the existing methods.

In [37], authors presented a transfer learning-based approach using a pretrained VGGNet architecture and images available in the training dataset, which consisted of 35,126 images in different classes. They used augmentation techniques to balance the dataset and achieved 96.61% accuracy for five classes of the same dataset. Bilal et al. [38] proposed a method based on two-stage feature extraction on the same dataset. In the first stage, they employed a pretrained U-Net-based transfer learning approach for feature extraction, and in the later stage, they used a novel CNN-SVD (Singular Value Decomposition) for deep learning features to classify the DR stages. They used a subset of the dataset that contained 7552 images in Class 0, 842 images in Class 1, 545 images in Class 2, 54 images in Class 3, and 95 images in Class 4. The best accuracy attained using this method was 97.92%.

Similarly, Luo et al. [39] suggested a different approach whereby they captured the global dependencies of images in the Kaggle EyePACS dataset. A correlation was found between the two input feature maps, and finally, the patch-wise information was embedded with the trained network for DR classification. As the dataset was imbalanced, they relied on F1 Score evaluation matric and achieved an 82.60% F1 Score. In addition, the accuracy of their proposed model was 83.60%.

Our proposed technique outperformed the aforementioned techniques in terms of classification performance, achieving an accuracy of 98.85%, sensitivity of 98.85%, specificity of 99.71%, precision of 98.89%, and an F1 Score of 98.85%. Table 9 compares the proposed technique with current state-of-the-art methods.

A graphical comparison of the proposed technique with existing techniques is shown in Figure 6. It is analyzed based on the fact that our method achieved improved classification results compared to the most recent research studies.

### 4.9. Quantitative Analysis of Proposed Method’s Average Performance

In this section, we discussed the experiments performed in terms of the average results. Table 10 provides a comparison of the outcomes of all classifiers using 5-fold cross-validation, and 50 epochs were used for the training of optimization algorithms. The findings indicate that the Quadratic SVM, which completed the task in 101.13 s, had the maximum detection and classification accuracy of 98.63%. Other performance measures achieved using the Quadratic SVM are SEN, PRE, SPE, and F1 scores of 98.63%, 98.67%, 99.66%, and 98.62% in 101.13 s.

When comparing the Fine Gaussian SVM to the other SVM classifiers, it achieved the poorest classification performance score, with an average accuracy of 41.70% in 1337.80 s. The confusion matrix in Figure 7a can be used to verify the Quadratic SVM results (performing better in 5-fold cross-validation than the other classifiers, and achieving the highest performance in this category) given in Figure 8a.

Similarly, Table 11 gives a comparison of the outcomes of all classifiers using 10-fold cross-validation, and 100 epochs were used for the training of optimization algorithms. The findings indicate that the Quadratic SVM, which completed the task in 180.88 s, had the maximum detection and classification accuracy of 98.85%. Other performance measures achieved using the Quadratic SVM are SEN, PRE, SPE, and F1 scores of 98.85%, 98.89%, 99.71%, and 98.85% in 180.88 s. These results can be verified using the confusion matrix of the Quadratic SVM classifier given in Figure 7b.

When comparing the Fine Gaussian SVM to the other SVM classifiers, it scored the poorest classification performance, with an average accuracy of 41.19% in 2042.40 s. The confusion matrix in Figure 7b can be used to verify the Quadratic SVM results of Experiment 3 given in Table 11. The top three classifiers are highlighted in the Table. Figure 8a,b show a visual depiction of the accuracies of all classifiers for the findings of Experiment 1 and Experiment 2. Similarly, Figure 9a,b provide a visual representation of the accuracies of all classifiers for the results of Experiment 3 and Experiment 4.

Upon analyzing the performance of the classifiers in terms of their achieved performance accuracy, it was found that the Quadratic SVM achieved the highest accuracy in all experiments. The maximum accuracy achieved in Experiment 3 using $F_{LBP}:15,000\times 30$, $F_{ResNet50}:15,000\times 500$, and $F_{GraphNet124}:15,000\times 500$ with 10-fold cross-validation was 98.85%.

## 5. Conclusions

Deep learning’s potential for detecting diabetic retinopathy has been illustrated in this study. With an optimized diabetic retinopathy dataset, we successfully identified diabetic retinopathy stages with the help of our proposed method based on deep convolutional neural networks. The findings of our study show that deep learning can be utilized for the classification of diabetic retinopathy into its five stages, thus offering healthcare practitioners a practical and affordable alternative. In conclusion, this study developed a hybrid technique that integrates image preprocessing with ensemble features for the computerized detection of diabetic retinopathy. Convolutional neural networks (CNNs) were utilized to create the model from scratch, fusing deep learning with local binary pattern (LBP) characteristics. The suggested model outperformed current state-of-the-art methods, achieving a high accuracy of 98.85%. The model could also distinguish between the proliferative and non-proliferative stages of DR with improved accuracy. The scope of our proposed hybrid model is limited to the detection and classification of diabetic retinopathy images only. It could also be applied to skin lesion detection, lungs cancer classification, mammographic image analysis, and other medical imaging related problems in the future. Specifically, this model can also be extended to diagnosing other retinal disorders, including glaucoma, age-related macular degeneration (AMD), and cataracts. Moreover, the model’s classification accuracy can be improved by utilizing statistical features with textural features in addition to the features extracted by the CNN.

## Figures and Tables

**Figure 1 diagnostics-13-01816-f001:**
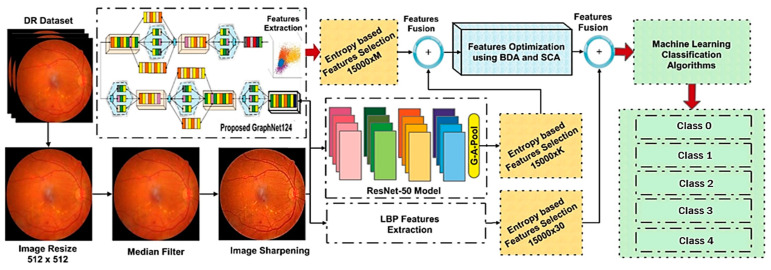
Proposed Model Block Diagram.

**Figure 2 diagnostics-13-01816-f002:**
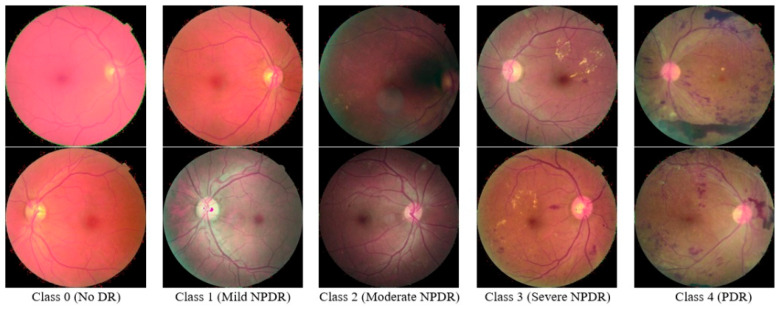
Sample Images of the Kaggle-EyePACS Dataset.

**Figure 3 diagnostics-13-01816-f003:**
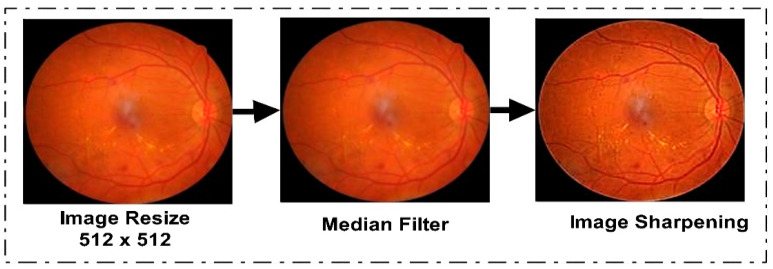
Proposed Dataset Preprocessing Phase Output.

**Figure 4 diagnostics-13-01816-f004:**
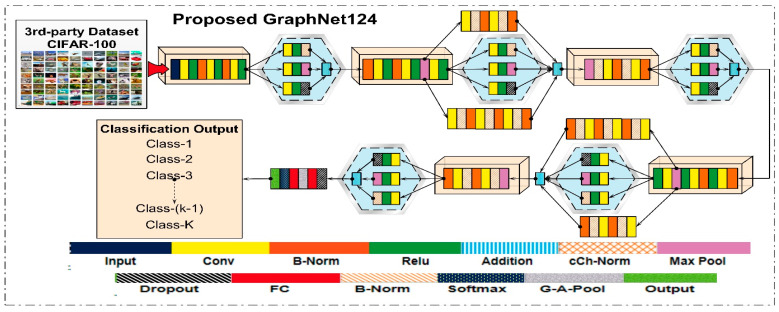
Proposed GraphNet-124 Architecture.

**Figure 5 diagnostics-13-01816-f005:**
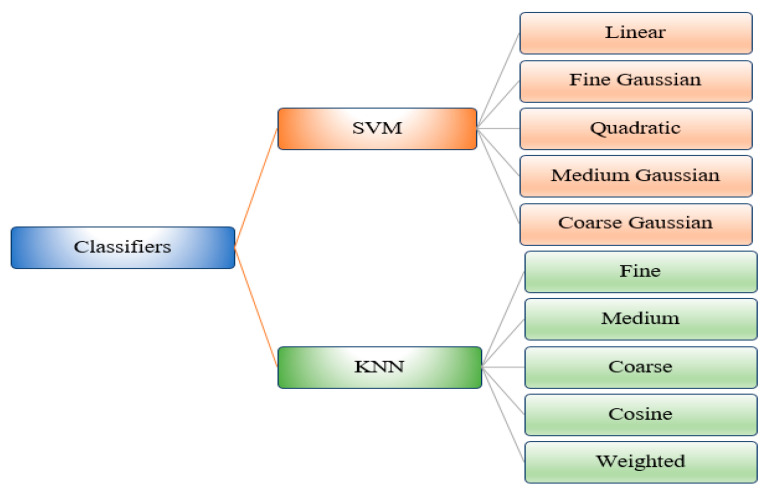
Hierarchal Representation of Classifiers.

**Figure 6 diagnostics-13-01816-f006:**
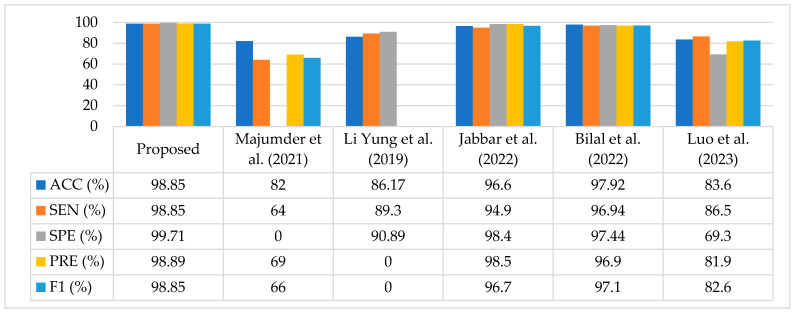
Analysis of the Proposed Approach Compared to Current Methods [35,36,37,38,39].

**Figure 7 diagnostics-13-01816-f007:**
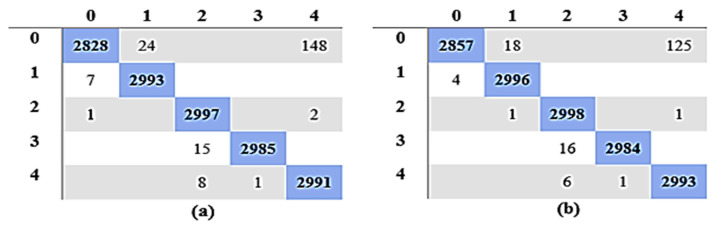
Confusion Matrix for Quadratic SVM: (**a**) Experiment 1; (**b**) Experiment 3.

**Figure 8 diagnostics-13-01816-f008:**
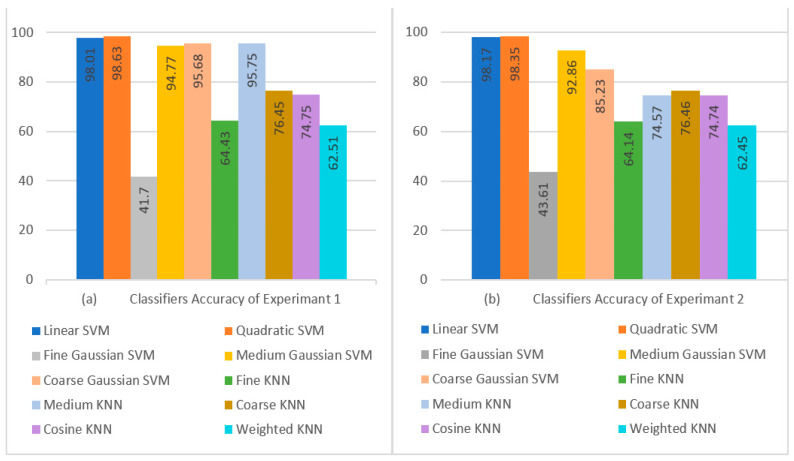
Visual Comparison of Classifier Performance: (**a**) Experiment 1 Results; (**b**) Experiment 2 Results.

**Figure 9 diagnostics-13-01816-f009:**
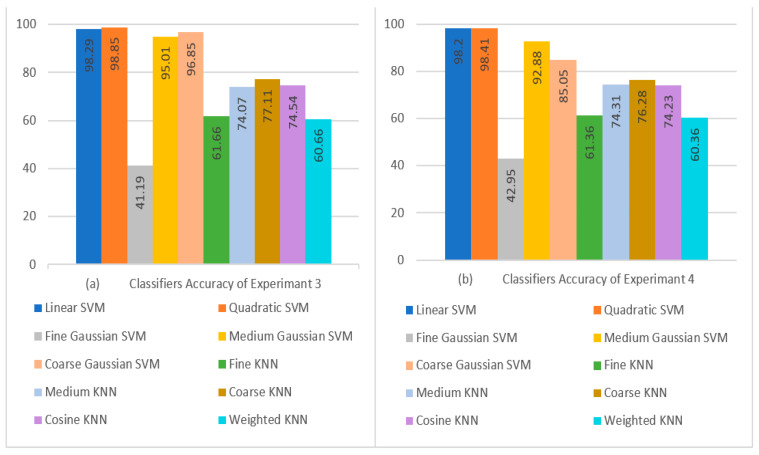
Visual Comparison of Classifier Performance: (**a**) Experiment 3 Results; (**b**) Experiment 4 Results.

**Table 1 diagnostics-13-01816-t001:** Class-Wise Quantitative Results of Experiment 1 Achieved using SVM Classifiers.

CLASSIFIER		CLASS	ACC (%)	SEN (%)	PRE (%)	SPE (%)	F1 (%)
**SVM**	Linear	0	98.01	93.10	99.43	99.87	96.16
		1		98.22	98.84	99.71	98.53
		2		99.73	98.97	99.74	99.35
		3		99.33	98.68	99.67	99.00
		4		99.63	94.38	98.52	96.94
	Quadratic	0	98.63	94.27	99.72	99.93	96.92
		1		99.77	99.20	99.80	99.48
		2		99.90	99.24	99.81	99.57
		3		99.50	99.97	99.99	99.73
		4		99.70	95.22	98.75	97.41
	Fine Gaussian	0	41.70	94.33	43.21	69.00	59.27
		1		6.630	7.480	79.48	7.030
		2		4.630	5.490	80.06	5.030
		3		54.07	100.00	100.00	70.19
		4		48.83	89.66	98.59	63.23
	Medium Gaussian	0	94.77	92.87	99.18	99.81	95.92
		1		90.87	90.26	97.55	90.56
		2		91.07	90.52	97.62	90.79
		3		99.27	99.93	99.98	99.60
		4		99.77	94.33	98.50	96.97
	Coarse Gaussian	0	95.68	91.17	98.10	99.56	94.51
		1		92.90	95.71	98.96	94.28
		2		97.90	91.32	97.68	94.50
		3		96.90	99.97	99.99	98.41
		4		99.53	94.02	98.42	96.70

**Table 2 diagnostics-13-01816-t002:** Class-Wise Quantitative Results of Experiment 1 Achieved using KNN Classifiers.

CLASSIFIER		CLASS	ACC (%)	SEN (%)	PRE (%)	SPE (%)	F1 (%)
**KNN**	Fine	0	64.43	91.47	92.83	98.23	92.14
		1		18.90	18.57	79.28	18.73
		2		18.60	18.53	79.55	18.56
		3		99.40	99.87	99.97	99.63
		4		93.80	94.02	98.51	93.91
	Medium	0	95.75	96.73	99.62	99.91	98.16
		1		90.03	95.14	98.85	92.52
		2		96.80	98.91	99.73	97.84
		3		98.70	93.41	98.26	95.98
		4		96.47	92.11	97.93	94.24
	Coarse	0	76.45	88.53	98.19	99.59	93.11
		1		48.57	47.93	86.81	48.25
		2		52.23	47.57	85.61	49.79
		3		94.57	100.00	100.00	97.21
		4		98.37	94.46	98.56	96.37
	Cosine	0	74.75	92.43	96.12	99.07	94.24
		1		63.87	44.38	79.99	52.37
		2		18.20	34.89	91.51	23.92
		3		99.30	99.37	99.84	99.33
		4		99.93	92.67	98.03	96.17
	Weighted	0	62.51	91.07	94.30	98.63	92.66
		1		13.90	13.61	77.94	13.75
		2		13.97	13.72	78.03	13.84
		3		98.87	100.00	100.00	99.43
		4		94.73	94.17	98.53	94.45

**Table 3 diagnostics-13-01816-t003:** Class-Wise Quantitative Results of Experiment 2 Achieved using SVM Classifiers.

CLASSIFIER		CLASS	ACC (%)	SEN (%)	PRE (%)	SPE (%)	F1 (%)
**SVM**	Linear	0	98.17	92.80	99.07	99.78	95.83
		1		99.43	98.58	99.64	99.00
		2		99.57	99.20	99.80	99.38
		3		99.33	99.97	99.99	99.65
		4		99.73	94.33	98.50	96.95
	Quadratic	0	98.35	93.10	99.54	99.89	96.21
		1		99.67	98.91	99.73	99.29
		2		99.73	99.30	99.83	99.52
		3		99.47	99.97	99.99	99.72
		4		99.77	94.33	98.50	96.97
	Fine Gaussian	0	43.61	85.57	50.82	79.30	63.77
		1		5.100	5.830	79.40	5.440
		2		3.530	4.210	79.92	3.840
		3		46.83	100.00	100.00	63.79
		4		77.03	67.91	90.90	72.18
	Medium Gaussian	0	92.86	92.87	99.15	99.80	95.90
		1		86.20	85.54	96.36	85.87
		2		86.17	85.88	96.46	86.02
		3		99.30	99.93	99.98	99.62
		4		99.77	94.24	98.48	96.92
	Coarse Gaussian	0	85.23	91.97	97.87	99.50	94.83
		1		47.17	80.58	97.16	59.50
		2		89.70	62.35	86.46	73.56
		3		97.67	100.00	100.00	98.82
		4		99.63	94.02	98.42	96.75

**Table 4 diagnostics-13-01816-t004:** Class-Wise Quantitative Results of Experiment 2 Achieved using KNN Classifiers.

CLASSIFIER		CLASS	ACC (%)	SEN (%)	PRE (%)	SPE (%)	F1 (%)
**KNN**	Fine	0	64.14	92.03	92.50	98.13	92.26
		1		18.03	17.79	79.17	17.91
		2		17.70	17.66	79.38	17.68
		3		99.63	99.90	99.98	99.77
		4		93.30	94.05	98.53	93.67
	Medium	0	75.57	91.57	97.83	99.49	94.59
		1		64.13	43.75	79.38	52.01
		2		18.57	33.64	90.84	23.93
		3		99.03	100.00	100.00	99.51
		4		99.57	94.32	98.50	96.87
	Coarse	0	76.46	89.37	98.13	99.58	93.55
		1		47.97	47.04	86.50	47.50
		2		50.13	47.18	85.97	48.61
		3		96.23	100.00	100.00	98.08
		4		98.60	94.38	98.53	96.45
	Cosine	0	74.74	93.00	95.65	98.94	94.30
		1		64.33	44.41	79.87	52.55
		2		16.87	33.96	91.80	22.54
		3		99.57	98.97	99.74	99.27
		4		99.93	92.85	98.08	96.26
	Weighted	0	62.45	91.57	94.50	98.67	93.01
		1		13.77	13.44	77.83	13.60
		2		12.67	12.62	78.07	12.64
		3		99.27	100.00	100.00	99.63
		4		94.97	94.03	98.49	94.49

**Table 5 diagnostics-13-01816-t005:** Class-Wise Quantitative Results of Experiment 3 Achieved using SVM Classifiers.

CLASSIFIER		CLASS	ACC (%)	SEN (%)	PRE (%)	SPE (%)	F1 (%)
**SVM**	Linear	0	98.29	93.07	99.36	99.85	96.11
		1		99.57	98.81	99.70	99.19
		2		99.67	99.17	99.79	99.42
		3		99.43	99.97	99.99	99.70
		4		99.70	94.41	98.53	96.98
	Quadratic	0	98.85	95.23	99.86	99.97	97.49
		1		99.87	99.37	99.84	99.62
		2		99.93	99.27	99.82	99.60
		3		99.47	99.97	99.99	99.72
		4		99.77	95.96	98.95	97.83
	Fine Gaussian	0	41.19	94.23	45.92	72.26	61.75
		1		4.200	4.470	77.56	4.330
		2		2.400	2.670	78.09	2.530
		3		56.33	100.00	100.00	72.07
		4		48.77	89.53	98.58	63.14
	Medium Gaussian	0	95.01	93.07	99.43	99.87	96.14
		1		91.47	90.56	97.62	91.01
		2		91.37	91.00	97.74	91.18
		3		99.40	100.00	100.00	99.70
		4		99.73	94.44	98.53	97.02
	Coarse Gaussian	0	96.85	91.23	98.03	99.54	94.51
		1		97.30	96.82	99.20	97.06
		2		98.93	95.74	98.90	97.31
		3		97.23	99.93	99.98	98.56
		4		99.57	94.11	98.44	96.76

**Table 6 diagnostics-13-01816-t006:** Class-Wise Quantitative Results of Experiment 3 Achieved using KNN Classifiers.

CLASSIFIER		CLASS	ACC (%)	SEN (%)	PRE (%)	SPE (%)	F1 (%)
**KNN**	Fine	0	61.66	91.87	93.20	98.33	92.53
		1		12.03	11.81	77.48	11.92
		2		11.33	11.30	77.78	11.32
		3		99.53	99.83	99.96	99.68
		4		93.70	94.04	98.52	93.87
	Medium	0	74.07	91.23	98.03	99.54	94.51
		1		69.67	43.34	77.23	53.44
		2		11.20	26.71	92.32	15.78
		3		98.63	100.00	100.00	99.31
		4		99.60	94.29	98.49	96.87
	Coarse	0	77.11	88.90	98.16	99.58	93.30
		1		50.03	49.25	87.10	49.64
		2		52.97	48.85	86.14	50.82
		3		95.07	100.00	100.00	97.47
		4		98.60	94.50	98.57	96.51
	Cosine	0	74.54	92.70	96.30	99.11	94.46
		1		69.97	44.39	78.08	54.31
		2		10.50	27.75	93.17	15.24
		3		99.53	99.24	99.81	99.38
		4		100.00	92.62	98.01	96.17
	Weighted	0	60.66	91.27	94.28	98.62	92.75
		1		9.830	9.520	76.63	9.670
		2		8.800	8.750	77.06	8.780
		3		98.97	100.00	100.00	99.48
		4		94.43	94.09	98.52	94.26

**Table 7 diagnostics-13-01816-t007:** Class-Wise Quantitative Results of Experiment 4 Achieved using SVM Classifiers.

CLASSIFIER		CLASS	ACC (%)	SEN (%)	PRE (%)	SPE (%)	F1 (%)
**SVM**	Linear	0	98.20	92.80	99.07	99.78	95.83
		1		99.47	98.61	99.65	99.04
		2		99.53	99.27	99.82	99.40
		3		99.43	99.97	99.99	99.70
		4		99.73	94.33	98.50	96.95
	Quadratic	0	98.41	93.23	99.50	99.88	96.27
		1		99.67	99.07	99.77	99.37
		2		99.77	99.40	99.85	99.58
		3		99.57	100.00	100.00	99.78
		4		99.80	94.36	98.51	97.00
	Fine Gaussian	0	42.95	90.33	51.83	79.35	65.86
		1		2.910	3.320	77.46	3.100
		2		2.000	2.060	76.58	2.030
		3		47.07	100.00	100.00	64.01
		4		75.10	79.22	95.16	77.10
	Medium Gaussian	0	92.88	92.17	99.35	99.85	95.63
		1		86.44	85.16	96.23	85.79
		2		86.67	86.24	96.54	86.45
		3		99.37	99.93	99.98	99.65
		4		99.77	94.30	98.49	96.95
	Coarse Gaussian	0	85.05	92.17	97.81	99.48	94.90
		1		41.20	86.43	98.38	55.80
		2		94.33	61.19	85.04	74.23
		3		97.77	99.90	99.98	98.82
		4		99.77	94.06	98.43	96.83

**Table 8 diagnostics-13-01816-t008:** Class-Wise Quantitative results of Experiment 4 Achieved using KNN Classifiers.

CLASSIFIER		CLASS	ACC (%)	SEN (%)	PRE (%)	SPE (%)	F1 (%)
**KNN**	Fine	0	61.36	92.53	93.63	98.43	93.08
		1		10.43	10.30	77.28	10.37
		2		10.27	10.23	77.48	10.25
		3		99.73	99.90	99.98	99.82
		4		93.83	94.12	98.53	93.97
	Medium	0	74.31	91.73	97.73	99.47	94.64
		1		70.07	43.66	77.40	53.80
		2		10.83	26.62	92.53	15.40
		3		99.27	99.93	99.98	99.60
		4		99.63	94.32	98.50	96.90
	Coarse	0	76.28	89.67	98.14	99.58	93.71
		1		47.13	46.41	86.39	46.77
		2		49.17	46.50	85.86	47.80
		3		96.60	100.00	100.00	98.27
		4		98.83	94.37	98.53	96.55
	Cosine	0	74.23	93.10	95.88	99.00	94.47
		1		68.70	43.94	78.08	53.60
		2		9.73	25.50	92.89	14.09
		3		99.60	98.81	99.70	99.20
		4		100.00	92.97	98.11	96.35
	Weighted	0	60.36	91.70	94.08	98.56	92.88
		1		8.40	8.200	76.49	8.300
		2		7.90	7.870	76.88	7.890
		3		99.43	99.93	99.98	99.68
		4		94.33	94.11	98.53	94.22

**Table 9 diagnostics-13-01816-t009:** Comparison of Proposed Method with Existing Techniques.

Ref.	Year	No. of Classes	Performance Measures				
			ACC (%)	SEN (%)	SPE (%)	PRE (%)	F1 (%)
[35]	2021	5	82.00	64.00	-	69.00	66.00
[36]	2019	5	86.17	89.30	90.89	-	-
[37]	2022	5	96.61	94.90	98.40	98.50	96.70
[38]	2022	5	97.92	96.94	97.44	96.90	97.10
[39]	2023	5	83.60	86.50	69.30	81.90	82.60
Proposed		5	98.85	98.85	99.71	98.89	98.85

**Table 10 diagnostics-13-01816-t010:** Comparison of Average Quantitative Results of Experiment 1 and Experiment 2.

CLASSIFIER		5-Fold Cross-Validation											
		Experiment 1						Experiment 2					
		$F_{LBP}:15,000\times 30$ $F_{ResNet50}:15,000\times 500$ $F_{GraphNet124}:15,000\times 500$						$F_{LBP}:15,000\times 30$ $F_{ResNet50}:15,000\times 1000$ $F_{GraphNet124}:15,000\times 1000$					
		ACC (%)	SEN (%)	PRE (%)	SPE (%)	F1 (%)	Time (s)	ACC (%)	SEN (%)	PRE (%)	SPE (%)	F1 (%)	Time (s)
**SVM**	Linear	98.01	98.00	98.06	99.50	98.00	81.62	98.17	98.17	98.23	99.54	98.17	190.21
	Quadratic	98.63	98.63	98.67	99.66	98.62	101.13	98.35	98.35	98.41	99.59	98.34	251.35
	Fine Gaussian	41.70	41.70	49.17	85.43	40.95	1337.80	43.61	43.61	45.75	85.90	41.81	2722.50
	Medium Gaussian	94.77	94.77	94.85	98.69	94.77	183.54	92.86	92.86	92.95	98.22	92.87	439.25
	Coarse Gaussian	95.68	95.68	95.82	98.92	95.68	212.74	85.23	85.23	86.97	96.31	84.69	431.17
**KNN**	Fine	64.43	64.43	64.76	91.11	64.60	223.19	64.14	64.14	64.38	91.04	64.26	432.57
	Medium	95.75	95.75	95.84	98.94	95.75	223.35	74.57	74.57	73.91	93.64	73.38	431.72
	Coarse	76.45	76.45	77.63	94.11	76.95	223.89	76.46	76.46	77.35	94.12	76.84	432.51
	Cosine	74.75	74.75	73.49	93.69	73.21	225.20	74.74	74.74	73.17	93.69	72.98	440.06
	Weighted	62.51	62.51	63.16	90.63	62.83	246.57	62.45	62.45	62.92	90.61	62.68	434.06

**Table 11 diagnostics-13-01816-t011:** Comparison of Average Quantitative Results of Experiment 3 and Experiment 4.

CLASSIFIER		10-Fold Cross-Validation											
		Experiment 3						Experiment 4					
		$F_{LBP}:15,000\times 30$ $F_{ResNet50}:15,000\times 500$ $F_{GraphNet124}:15,000\times 500$						$F_{LBP}:15,000\times 30$ $F_{ResNet50}:15,000\times 1000$ $F_{GraphNet124}:15,000\times 1000$					
		ACC (%)	SEN (%)	PRE (%)	SPE (%)	F1 (%)	Time (s)	ACC (%)	SEN (%)	PRE (%)	SPE (%)	F1 (%)	Time (s)
**SVM**	Linear	98.29	98.29	98.34	99.57	98.28	169.26	98.20	98.19	98.25	99.55	98.19	418.31
	Quadratic	98.85	98.85	98.89	99.71	98.85	180.88	98.41	98.41	98.47	99.60	98.40	470.49
	Fine Gaussian	41.19	41.19	48.52	85.30	40.76	2042.40	42.95	43.48	47.29	85.71	42.42	5534.27
	Medium Gaussian	95.01	95.01	95.09	98.75	95.01	301.35	92.88	92.88	93.00	98.22	92.89	799.76
	Coarse Gaussian	96.85	96.85	96.93	99.21	96.84	346.30	85.05	85.05	87.88	96.26	84.12	793.14
**KNN**	Fine	61.66	61.69	62.04	90.41	61.86	270.30	61.36	61.36	61.63	90.34	61.50	1087.80
	Medium	74.07	74.07	72.47	93.52	71.98	273.56	74.31	74.31	72.45	93.58	72.07	1759.80
	Coarse	77.11	77.11	78.15	94.28	77.55	257.45	76.28	76.28	77.08	94.07	76.62	1557.70
	Cosine	74.54	74.54	72.06	93.64	71.91	252.98	74.23	74.23	71.42	93.56	71.54	1545.30
	Weighted	60.66	60.66	61.33	90.17	60.99	247.85	60.36	60.35	60.84	90.09	60.59	480.97

## Data Availability

The dataset utilized for conducting this research was acquired from the publicly available Kaggle EyePACS [21] database.

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
