# Peer review of "A Hybrid Technique for Diabetic Retinopathy Detection Based on Ensemble-Optimized CNN and Texture Features"

_diagnostics, 2023, doi:10.3390/diagnostics13101816_

Round 1

Reviewer 1 Report

This paper presents a method for detecting and classifying DR abnormalities by using a hybrid approach based on image preprocessing and ensemble features. The approach combines Local Binary Patterns (LBP) with deep learning features to create the ensemble feature vector, which is then used to train machine learning classifiers. However, the contribution of this paper seems limited, as the authors only combined LBP and deep learning features and utilized SVM as the machine learning algorithm to detect the severity level.

Moreover, the paper contains numerous grammatical errors and is poorly written. The overuse of the statement "Error! Reference source not found" indicates a lack of careful revision, which is concerning. These issues may negatively impact the reader's understanding of the paper's content and its contributions to the field.

In addition, the paper lacks a clear explanation of the methodology, making it difficult for readers to understand how the proposed approach works. The authors could have improved the clarity of their methodology by providing a step-by-step description of their process. Furthermore, the paper fails to present any comparative analysis of the proposed approach with other state-of-the-art techniques, which raises concerns about the effectiveness of the method.

Overall, the paper could benefit from extensive revisions to improve its language, structure, and technical content. The authors should consider seeking feedback from subject matter experts to strengthen the paper's contributions and significance in the field of DR detection and classification.

Author Response

Response to Reviewer 1 Comments

(Manuscript No. diagnostics-2343208)

Title: A Hybrid Technique for Diabetic Retinopathy Detection based on Ensemble Optimized CNN and Texture Features

Dear Editor,

We appreciate the time and effort exerted by the editor and referees in reviewing this manuscript. We are very thankful to the reviewers for their deep review and accepting our work with major revisions. Our manuscript is revised considering their useful suggestions and comments. Our responses to their specific comments/suggestions/queries are presented below.  

Reviewer-I Responses

Comment 1.1: This paper presents a method for detecting and classifying DR abnormalities by using a hybrid approach based on image preprocessing and ensemble features. The approach combines Local Binary Patterns (LBP) with deep learning features to create the ensemble feature vector, which is then used to train machine learning classifiers. However, the contribution of this paper seems limited, as the authors only combined LBP and deep learning features and utilized SVM as the machine learning algorithm to detect the severity level.

Response 1.1: Thank you for your understanding of our work that we employed a hybrid approach based on image preprocessing and creating ensemble feature vector based on two convolutional neural networks (including, the novel GraphNet124 which was created from scratch and ResNet50 which was utilized as a transfer learning network) and LBP (textural features). Afterwards, the features were fused. To the best of our knowledge, this study is the first in the domain of DR abnormalities’ detection and classification using fusion of a novel automated CNN (GraphNet124) and transfer learning (ResNet50) based features and LBP-based textural features. Later, we utilized 10 Machine Learning (ML) algorithms, including five variants of SVM (Linear, Quadratic, Fine Gaussian, Medium Gaussian, Coarse Gaussian) and five variants of KNN (Fine, Medium, Coarse, Cosine, Weighted) for evaluation of our proposed hybrid model. Quadratic SVM showed promising performance with highest accuracy of 98.85%.

Comment 1.2: Moreover, the paper contains numerous grammatical errors and is poorly written. The overuse of the statement "Error! Reference source not found" indicates a lack of careful revision, which is concerning. These issues may negatively impact the reader's understanding of the paper's content and its contributions to the field.

Response 1.2: Thank you for pointing this out. In the revised manuscript, we have polished the paper according to the reviewer’s suggestion. Furthermore, we have corrected all of the broken references as indicated by the reviewer in Section 3, Section 3.1, Section 3.2, Section 3.3.2, Section 4, Section 4.4, Section 4.5, Section 4.6, Section 4.7, Section 4.8, Section 4.9.

Comments 1.3: In addition, the paper lacks a clear explanation of the methodology, making it difficult for readers to understand how the proposed approach works. The authors could have improved the clarity of their methodology by providing a step-by-step description of their process.

Response 1.3: Thank you for your comment and valuable suggestion. In the revised manuscript, we have provided step-by-step description of our proposed methodology as advised by the reviewer. In the revised manuscript, a clear explanation of our methodology is described in Section 3. It will be helpful for the readers to fully understand our proposed work.

Comment 1.4: Furthermore, the paper fails to present any comparative analysis of the proposed approach with other state-of-the-art techniques, which raises concerns about the effectiveness of the method.

Response 1.4: Thank you for the comment. The comparative analysis is present in Section 4.8. Moreover, we have rewritten this section, discussed the state-of-the-art techniques in detail and compared our proposed model with them in our revised manuscript. We have discussed their proposed methodologies and the accuracies achieved by these models. Furthermore, Table 9 of our revised manuscript shows the comparison of our proposed method with existing state-of-the-art techniques in terms of Accuracy, Sensitivity, Specificity, Precision and F1-Score.

Comment 1.5: Overall, the paper could benefit from extensive revisions to improve its language, structure, and technical content. The authors should consider seeking feedback from subject matter experts to strengthen the paper's contributions and significance in the field of DR detection and classification.

Response 1.5: Thank you for the comment. We have revised our manuscript in accordance with the valuable suggestions by the reviewer. We have improved the language of our manuscript by involving a native English speaker. The structure of our manuscript is improved by following the effective comments by the reviewer. Moreover, for technical content, we improved it by requesting a subject area expert who provided important feedback on our proposed methodology. The paper’s contributions and significance are already mentioned in the manuscript in Section 1, that is, Introduction. Furthermore, for the clear understanding of our manuscript, we have elaborated our methodology in Section 3 which includes the following:

  1. Dataset: An online public dataset, Kaggle- EyePACS dataset was used for the detection and classification of DR images into its specific classes.
  2. Preprocessing: The DR images were then preprocessed since preprocessing is an important phase in DR detection and classification. The preprocessing steps followed in this study were image resizing, data augmentation, applying median filter and image sharpening.
  3. Features Engineering: Distinguishing features were then extracted and selected from the preprocessed dataset. Three methods were used for feature extraction, namely, Local Binary Patterns (LBP) for texture-oriented features, the novel GraphNet124 and ResNet50 for CNN-based features.
  4. Features Selection and Fusion: After feature extraction, salient features from LBP, GraphNet124 and ResNet50 were selected using Shannon Entropy algorithm. Moreover, these selected features were then fused and optimized using the Binary Dragonfly Algorithm (BDA), and Sine Cosine Algorithm (SCA).
  5. Classification and Evaluation: The optimized feature vector was fed to ten ML algorithms, including five variants on SVM and five variants of KNN for classification of DR images into its five severity classes. Finally, these algorithms were evaluated on different evaluation matrices, namely, Specificity (SPE), F1-Score (F1), Accuracy (ACC), Precision (PRE), Sensitivity (SEN), and Time (in Seconds).

Reviewer 2 Report

1.     The literature review is just a pile of information, lacking of analysis and induction.

2.     Some English structures are challenging to understand.

3.     In 169,186,196,271,329, etc, change  Error! Reference source not found” using reference

4.     In figure 1, change class 3 by Class 4

5.     Please formatted the paragraphs and tables.

6.     In 197, explain why the authors employed the median filter on the entire dataset.

7.     The figures are not mentioned in the paragraphs.

8.      The authors should have explain the choice of hyperparameters in Deep learning model.

9.     The discussion section in the present form is relatively weak and should be strengthened with more details and justifications.

10.  In conclusion, more discussions should be made regarding the limitation and future dimensions of the proposed study.

Author Response

Response to Reviewer 2 Comments

(Manuscript No. diagnostics-2343208)

Title: A Hybrid Technique for Diabetic Retinopathy Detection based on Ensemble Optimized CNN and Texture Features

Dear Editor,

We appreciate the time and effort exerted by the editor and referees in reviewing this manuscript. We are very thankful to the reviewers for their deep review and accepting our work with major revisions. Our manuscript is revised considering their useful suggestions and comments. Our responses to their specific comments/suggestions/queries are presented below.  

Reviewer-II Responses

Comment 2.1: The literature review is just a pile of information, lacking of analysis and induction.

Response 2.1: Thank you for your valuable comment. In the revised manuscript, for literature review, we have rewritten the Related Work Section as per given suggestion. Moreover, the updated Related Work Section clearly analysis the currently available literature and defines the rationale for the proposed methodology. The updated Related Work can be found in Section 2.

Comment 2.2: Some English structures are challenging to understand.

Response 2.2: Thank you for the comment. We have revised our manuscript in accordance with the valuable suggestions by the reviewer. We have improved the language of our manuscript by involving a native English speaker.

Comment 2.3: In 169,186,196,271,329, etc, change ” Error! Reference source not found” using reference.

Response 2.3: Thank you for pointing this out. In the revised manuscript, we have polished the paper according to the reviewer’s suggestion. Furthermore, we have corrected all of the broken references as indicated by the reviewer in Section 3, Section 3.1, Section 3.2, Section 3.3.2, Section 4, Section 4.4, Section 4.5, Section 4.6, Section 4.7, Section 4.8, Section 4.9.

Comment 2.4: In figure 1, change class 3 by Class 4.

Response 2.4: Thank you for pointing this out. Class 3 was mistakenly written twice in Figure 1. In our revised manuscript, we have corrected this, and the five classes of Diabetic Retinopathy severity levels are written accordingly as, Class 0, Class 1, Class 2, Class 3, and Class 4.

Comment 2.5: Please formatted the paragraphs and tables.

Response 2.5: Thank you for the comment. The revised manuscript has been updated as per given suggestion. In the revised manuscript, we have formatted all the paragraphs and tables properly and the broken references of the Tables are updated in Section 4.4 Table 1, Section 4.4 Table 2, Section 4.5 Table 3, Section 4.5 Table 4, Section 4.6 Table 5, Section 4.6 Table 6, Section 4.7 Table 7, Section 4.7 Table 8, Section 4.8 Table 9, Section 4.9 Table 10, Section 4.9 Table 11.  

Comment 2.6: In 197, explain why the authors employed the median filter on the entire dataset.

Response 2.6: Thank you for your comment. We employed the median filter on the entire dataset for noise removal from the images since median filter is an image smoothing technique and it retains the edges while noise removal. Figure 3 of our manuscript is showing the effect of median filtering. In our revised manuscript, we have clearly mentioned the reason of employing median filter on the dataset in Section 3.2.

Comment 2.7: The figures are not mentioned in the paragraphs.

Response 2.7: Thank you. In the revised manuscript, we have updated the broken references of the figures and mentioned all the figures, e.g., Section 3 Figure 1, Section 3.1 Figure 2, Section 3.2 Figure 3, Section 3.3.2 Figure 4, Section 4 Figure 5, Section 4.8 Figure 6, Section 4.9 Figure 7a and Figure 7b, Section 4.9 Figure 8a and Figure 8b, Section 4.9 Figure 9a and Figure 9b.

Comment 2.8: The authors should have explain the choice of hyperparameters in Deep learning model.

Response 2.8: Thank you for highlighting this important issue as the choice of hyperparameters must be mentioned that will surely be beneficial for the readers. In our revised manuscript, we have discussed in detail regarding the hyperparameters utilized in Section 3.3.2. We have described in our revised manuscript that; training of our deep neural network was performed using the process of fine-tuning the hyperparameters. Moreover, we trained the model using SGDM (Stochastic Gradient Descent with momentum) optimizer, with the validation frequency of 50, and the maximum epochs used for the training were 50 and 100 for 5-fold and 10-fold Cross-Validation experiments respectively, with minibatch size of 64. Furthermore, we utilized L2 regularization of 0.0001 and shuffled images at every epoch with the learning rate dropped by the factor 0.1.  

Comment 2.9: The discussion section in the present form is relatively weak and should be strengthened with more details and justifications.

Response 2.9: Thank you for your comment regarding discussion part that is present in Section 4 Results and Discussion. In our revised manuscript, we have rewritten the Section 4.8 Comparison with Existing Methods with more details and justification in our revised manuscript. For instance, “Authors in [34], proposed a method for the DR recognition for the EyePACS dataset. This model achieved a promising accuracy of 82%.” is rewritten in detail and justifications are provided regarding their model and the number of images used by them to train their model, as follows:

“Authors in [34], proposed a method for the DR recognition, named, MXception model for the Kaggle EyePACS dataset. They trained their model on a subset of the dataset comprising a total 19,316 images (10,000 Class 0, 2,443 Class 1, 5,292 Class 2, 873 Class 3, and 708 Class 4). To make the dataset balanced, they used a class weighting method. Moreover, they employed a pretrained Xception model by chopping the last fully connected layer and added an average pooling layer and a one neuron dense layer as their output layer. This model achieved a promising accuracy of 82%.”

Similarly, “Bilal et al. [37] proposed a method based on deep learning features for DR classification. The best accuracy attained by this method was 97.92%.” is deliberated in detail and justified as follows:

“Bilal et al. [37] proposed a method based on two stage features extraction on the same dataset. In the first stage, they employed a pretrained U-Net based transfer learning approach for feature extraction and in the later stage they used a novel CNN-SVD (Singular Value Decomposition) for deep learning features to classify DR stages. They used a subset of the dataset containing 7552 images in Class 0, 842 images in Class 1, 545 images in Class 2, 54 images in Class 3 and 95 images in Class 4. The best accuracy attained by this method was 97.92%.”

Moreover, other state-of-the-art models are also discussed in detail and justifications are provided accordingly in the light of the valuable suggestions by the reviewer.

Comment 2.10: In conclusion, more discussions should be made regarding the limitation and future dimensions of the proposed study.

Response 2.10: Thank you for the suggestion. We have discussed the limitations of our proposed study as well as the future dimensions for the researchers working in the domain of retinal disorders. We have updated the Conclusions Section of our revised manuscript in accordance with the valuable comment by the reviewer.

Reviewer 3 Report

The authors proposed a hybrid deep-learning based framework for automated diabetic retinopathy (DR) detection. The features extracted by deep neural network and LBP features are fused to improve the classification accuracy. Evaluation was done on a public data set.

Major comments:

1. Severe reference errors exist, e.g., Line 169, 186, 196, 238, 272, 329, 351, 358, 382, 391, 415, 416, 424, 434, 439, 446, 456, 458, 472-479, making the whole manuscript difficult to read.

2. In chapter 4.8, the comparison with existing methods lack evaluation details. Did the authors directly use the experiment results from those publications, or implement those methods and test on the same dataset as the proposed method? If it’s the former, please specify the difference of the data sets. If it’s the latter, please give the implementation details of these methods.

Minor comments:

1. In Figure 9, the maximum Accuracy should be set to 100.

2. Please increase the resolution of Figure 8.

3. Please check the fonts in Tables, some are not able to be seen, e.g., Table 1.

4. Line 465: wrong Table index.

5. Font size inconsistency in Line 125-128.

6. Two ‘Class 3’ in Figure 1.

7. For Figure 2, it’s better to show samples from different classes.

8. All the equations should be numbered.

9. The equation in Line 222: I should be written as superscript. And should S() be D(), as shown in Line 226?

Author Response

Response to Reviewer 3 Comments

(Manuscript No. diagnostics-2343208)

Title: A Hybrid Technique for Diabetic Retinopathy Detection based on Ensemble Optimized CNN and Texture Features

Dear Editor,

We appreciate the time and effort exerted by the editor and referees in reviewing this manuscript. We are very thankful to the reviewers for their deep review and accepting our work with major revisions. Our manuscript is revised considering their useful suggestions and comments. Our responses to their specific comments/suggestions/queries are presented below.  

Reviewer-III Responses

Major Comments / Responses

Major Comment 3.1: Severe reference errors exist, e.g., Line 169, 186, 196, 238, 272, 329, 351, 358, 382, 391, 415, 416, 424, 434, 439, 446, 456, 458, 472-479, making the whole manuscript difficult to read.

Major Response 3.1: Thank you for pointing this out. In the revised manuscript, we have polished the paper according to the reviewer’s suggestion. Furthermore, we have corrected the broken references as indicated by the reviewer in Section 3, Section 3.1, Section 3.2, Section 3.3.2, Section 4, Section 4.4, Section 4.5, Section 4.6, Section 4.7, Section 4.8, Section 4.9.

Major Comment 3.2: In chapter 4.8, the comparison with existing methods lack evaluation details. Did the authors directly use the experiment results from those publications, or implement those methods and test on the same dataset as the proposed method? If it’s the former, please specify the difference of the data sets. If it’s the latter, please give the implementation details of these methods.

Major Response 3.2: Thank you for highlighting the comparison details issue. In our revised manuscript, since we had directly used the experimental results from publications, therefore, we have discussed the number of images from the same dataset (i.e. Kaggle EyePACS) utilized by their authors. We have rewritten the Section 4.8 Comparison with Existing Methods in our revised manuscript. For instance, “Authors in [34], proposed a method for the DR recognition for the EyePACS dataset. This model achieved a promising accuracy of 82%.” is rewritten in detail and justifications are provided regarding their model and the number of images used by them to train their model, as follows:

“Authors in [34], proposed a method for the DR recognition, named, MXception model for the Kaggle EyePACS dataset. They trained their model on a subset of the dataset comprising a total 19,316 images (10,000 Class 0, 2,443 Class 1, 5,292 Class 2, 873 Class 3, and 708 Class 4). To make the dataset balanced, they used a class weighting method. Moreover, they employed a pretrained Xception model by chopping the last fully connected layer and added an average pooling layer and a one neuron dense layer as their output layer. This model achieved a promising accuracy of 82%.”

Similarly, “Bilal et al. [37] proposed a method based on deep learning features for DR classification. The best accuracy attained by this method was 97.92%.” is deliberated in detail and justified as follows:

“Bilal et al. [37] proposed a method based on two stage features extraction on the same dataset. In the first stage, they employed a pretrained U-Net based transfer learning approach for feature extraction and in the later stage they used a novel CNN-SVD (Singular Value Decomposition) for deep learning features to classify DR stages. They used a subset of the dataset containing 7552 images in Class 0, 842 images in Class 1, 545 images in Class 2, 54 images in Class 3 and 95 images in Class 4. The best accuracy attained by this method was 97.92%.”

Moreover, other state-of-the-art models are also discussed in detail and justifications are provided accordingly in the light of the valuable suggestions by the reviewer.

Minor Comments / Responses

Comment 3.1: In Figure 9, the maximum Accuracy should be set to 100.

Response 3.1: Thank you for pointing this out. In the revised manuscript, we have corrected this, and the maximum accuracy is set to 100 in Figure 9. In addition, when we were revising our manuscript, we found the maximum accuracy in Figure 8 was also 120 mistakenly. So, we corrected it to be 100.

Comment 3.2: Please increase the resolution of Figure 8.

Response 3.2: Thank you for the comment. We increased the resolution of Figure 8 and similarly of Figure 9, but still they were looking blurred. Therefore, keeping in view your valuable suggestion, we recreated Figure 8 and Figure 9. Both the versions of Figure 8 (one from our old manuscript and the one from our new manuscript) are listed below:

*See the attachment*

Similarly, we improved Figure 9 of our revised manuscript in the same manner.

Comment 3.3: Please check the fonts in Tables, some are not able to be seen, e.g., Table 1.

Response 3.3: Thank you for the comment. The revised manuscript has been updated as per given suggestion. In the revised manuscript, we have formatted fonts in Tables properly and now they can easily be seen. For instance, when the complete phrase “Support Vector Machine Classifiers” was used in Section 4.4 Table 1, it could not be seen properly in the Journal’s format. So, this complete phrase is replaced with “SVM”. Moreover, when the complete phrase “K-Nearest Neighbors Classifiers” was used in Section 4.4 Table 2, again it could not be read properly in the Journal’s format. So, this phrase is replaced with KNN in our revised manuscript. Similarly, the rest of the Tables in the manuscript are also updated.

Comment 3.4: Line 465: wrong Table index.

Response 3.4: Thank you for mentioning this wrong Table index. In the revised manuscript, we have updated the Table 11 index.

Comment 3.5: Font size inconsistency in Line 125-128.

Response 3.5: Thank you. The Font size inconsistency has been removed and the Fonts in whole revised manuscript are consistent.

Comment 3.6: Two ‘Class 3’ in Figure 1.

Response 3.6: Thank you for highlighting this. Class 3 was mistakenly written twice in Figure 1. In our revised manuscript, we have corrected this, and the five classes of Diabetic Retinopathy severity levels are written accordingly as, Class 0, Class 1, Class 2, Class 3, and Class 4.

Comment 3.7: For Figure 2, it’s better to show samples from different classes.

Response 3.7: Thank you for the valuable suggestion. In our revised manuscript, we have shown samples from all 5 classes with proper captioning. Following is the Figure 2 used in our revised manuscript:

*See the attachment*

Figure 2: Sample Images of the Kaggle-EyePACS Dataset

Comment 3.8: All the equations should be numbered.

Response 3.8: Thank you for the comment. First two equations in Section 3.3.1 were left unnumbered mistakenly. In our revised manuscript, we have numbered these two equations and the numbers of other equations in Section 3.3.3 and Section 4.3 are updated accordingly.

Comment 3.9: The equation in Line 222: I should be written as superscript. And should S() be D(), as shown in Line 226?

Response 3.9: Thank you for pointing this out. In the revised manuscript, we have updated the equation (1) and made  the superscript as . Moreover, as indicated by the reviewer, here () should be () as it is representing the surrounding pixels. Hence, equation (2) is also updated accordingly.

Round 2

Reviewer 3 Report

The authors proposed a hybrid deep-learning based framework for automated diabetic retinopathy (DR) detection. The features extracted by deep neural network and LBP features are fused to improve the classification accuracy. Evaluation was done on a public data set, making it possible to compare with other existing methods.

Author Response

Response to Reviewer 3 Comments (Round 2)

 (Manuscript No. diagnostics-2343208)

Title: A Hybrid Technique for Diabetic Retinopathy Detection based on Ensemble Optimized CNN and Texture Features

Dear Editor,

We appreciate the time and effort exerted by the editor and referees in reviewing this manuscript. We are very thankful to the reviewers for their deep review and accepting our work with revisions. Our manuscript is revised considering their useful suggestions and comments. Our responses to their specific comments/suggestions/queries are presented below.  

Reviewer-III Responses (Round 2)

Comment 3.1: I've found that the manuscript is not professionally prepared. For example, numbering of figures are not in order (a, b,..) and there is a duplicated figure (even it's cropped).

Response 3.1: Thank you for pointing this out. In the revised manuscript, we have polished the paper according to the reviewer’s suggestion. For instance, the numbering of Figure 8(a), 8(b) and Figure 9(a), 9(b) are in order. These figures are listed below:

* Uploaded in the file named, "Response to Reviewer Comments 3 Round 2"

Figure 8: Visual Comparison of Classifiers Performance (a) Experiment 1 Results (b) Experiment 2 Results

* Uploaded in the file named, "Response to Reviewer Comments 3 Round 2"

Figure 9: Visual Comparison of Classifiers Performance (a) Experiment 3 Results (b) Experiment 4 Results

Moreover, Figure 1 was duplicated and shown cropped because of cross-referencing issue. Now, we have resolved this issue and the figure is shown at its proper place and there in no duplicated and cropped figure.

Comment 3.2: In addition, academic writing should be prepared in an appropriate manner including the figures and tables as well.

Response 3.2: Thank you for your comment. We have updated our manuscript according to the reviewer’s suggestions. We have thoroughly and rigorously rewritten it as an academic piece (changes can be tracked in the submitted manuscript). Moreover, the figures and tables are also updated according to the reviewer’s recommendations. For instance, some sample images of different classes of Diabetic Retinopathy in Figure 2 were overlapped and seemed cropped. In our revised manuscript, we improved Figure 2. Furthermore, Figure 8 and Figure 9 are also updated. Additionally, the line-spacings of Table 1 through Table 8 are also adjusted in our revised manuscript. We tried our best to manage each table on the same page where it started, as it presents our findings in a better way for the readers’ understanding.
